# Carbon-Based Nanocomposites as Fenton-Like Catalysts in Wastewater Treatment Applications: A Review

**DOI:** 10.3390/ma14102643

**Published:** 2021-05-18

**Authors:** Ling Xin, Jiwei Hu, Yiqiu Xiang, Caifang Li, Liya Fu, Qiuhua Li, Xionghui Wei

**Affiliations:** 1Guizhou Provincial Key Laboratory for Information Systems of Mountainous Areas and Protection of Ecological Environment, Guizhou Normal University, Guiyang 550001, China; xinling901231@163.com (L.X.); xxiangyiqiu@163.com (Y.X.); lcflicaifang@163.com (C.L.); qiuhua2002@126.com (Q.L.); 2Cultivation Base of Guizhou National Key Laboratory of Mountainous Karst Eco-Environment, Institute of Karst, Guizhou Normal University, Guiyang 550001, China; 3Research Center of Water Pollution Control Technology, Chinese Research Academy of Environmental Sciences, Chaoyang District, Beijing 100012, China; fuliya1986@126.com; 4Guizhou International Science and Technology Cooperation Base-International Joint Research Centre for Aquatic Ecology, Guizhou Normal University, Guiyang 550001, China; 5Department of Applied Chemistry, College of Chemistry and Molecular Engineering, Peking University, Beijing 100871, China; xhwei@pku.edu.cn

**Keywords:** hydrogen peroxide, degradation, fenton process, organic pollutants, free radicals

## Abstract

Advanced oxidation (e.g., fenton-like reagent oxidation and ozone oxidation) is a highly important technology that uses strong oxidizing free radicals to degrade organic pollutants and mineralize them. The fenton-like reactions have the characteristics of low cost, simple operation, thorough reaction and no secondary pollution. Fenton-like reagents refer to a strong oxidation system composed of transition metal ions (e.g., Fe^3+^, Mn^2+^ and Ag^+^) and oxidants (hydrogen peroxide, potassium persulfate, sodium persulfate, etc). Graphene and carbon nanotube possess a distinctive mechanical strength, flexibility, electrical and thermal conductivity and a very large specific surface area, which can work as an excellent carrier to disperse the catalyst and prevent its agglomeration. Fullerene can synergize with iron-based materials to promote the reaction of hydroxyl groups with organic pollutants and enhance the catalytic effect. Fenton-like catalysts influence the catalytic behavior by inducing electron transfer under strong interactions with the support. Due to the short lifespan of free radicals, the treatment effect is usually enhanced with the assistance of external conditions (ultraviolet and electric fields) to expand the application of fenton-like catalysts in water treatment. There are mainly light-fenton, electro-fenton and photoelectric-fenton methods. Fenton-like catalysts can be prepared by hydrothermal method, impregnation and coordination-precipitation approaches. The structures and properties of the catalysts are characterized by a variety of techniques, such as high-resolution transmission electron microscopy, high-angle annular dark-field scanning transmission electron microscopy and X-ray absorption near-edge structure spectroscopy. In this paper, we review the mechanisms, preparation methods, characterizations and applications status of fenton-like reagents in industrial wastewater treatment, and summarize the recycling of these catalysts and describe prospects for their future research directions.

## 1. Introduction

Water is the source of life, and water quality has a direct impact on our survival and development. Today’s economy has been developing rapidly and people’s living standards have been improving. However, environmental problems are becoming gradually prominent, and there are increasing types of industrial and agricultural wastewater as well as urban sewage [1,2]. Water pollution not only aggravates the increasingly serious problem of water shortage, but also poses a great threat to human health [3,4,5]. The main sources of water pollution are: physical, chemical and biological pollution [6,7]. Among them, chemical pollution is highly prevalent in our society, mainly referring to industrial wastewater containing various toxic contaminants (such as dyes, phenolic compounds and polycyclic aromatic hydrocarbons) and urban sewage containing pharmaceutical and personal care products, and agricultural wastewater contaminated with pesticides, insecticides and herbicides, etc., which are released without treatment [8,9,10]. Pharmaceutical and personal care products (PPCPs) are emerging pollutants that have a profound and irreversible influence on human health despite their low concentrations in the environment. Polycyclic aromatic hydrocarbons (PAHs) exist frequently in the form of complex mixtures in water and have strong bioaccumulation and persistence, and have been shown in animal studies to cause a carcinogenic effect, thus belonging to the priority pollutants. Dyes are colored organic compounds, which are resistant to photolysis, oxidation and biodegradation, and make them extremely difficult to handle with general water treatment systems. Hence, a deep treatment is necessary before discharging wastewater to make effluent emissions standards more stringent and to achieve the objectives of improving water quality and protecting the ecological environment.

At present, relatively mature wastewater treatment methods are mainly divided into biological technology, physical approaches, chemical processes and advanced oxidation methods [11,12,13]. Among them, ozone oxidation is an advanced oxidation method where hydroxyl radicals can be generated by catalytic activation of O_3_ through catalysts (e.g., Fe^2+^, Mn^2+^, Ni^2+^, Cu^2+^, Ag^+^ and Zn^2+^) or solid catalysts (metal oxides, porous materials and their composites) [14,15,16]. Chen et al. [17] examined the operation of O_3_/H_2_O_2_ catalytic system for the treatment of semi-coking wastewater and found its excellent degradation performance, easy magnetic separation and high stability. This proved that this system provided a theoretical basis for the deep treatment of coal chemical wastewater by ozone oxidation and had good practical application prospects and values. Besides, O_3_ catalytic oxidation technology is simple to implement and easy to manage, which can achieve the zero emissions [18]. 

The fenton oxidation reaction is simple to operate and can better oxidize and degrade refractory organics in the wastewater, but there are problems such as insufficient mineralization of the organics, secondary pollution caused by a large amount of iron sludge and low utilization of H_2_O_2_ [19]. To solve these problems, many fenton-like techniques have been developed based on the conventional fenton method, such as light-fenton, electrical-fenton, and ultrasonic-fenton methods [20]. Fenton-like reagents are composed of oxidants and transition metal ions (Fe^3+^, Cu^2+^, Mn^2+^, Ni^2+^, etc.), iron-containing minerals, and metal-filled polymers [20,21]. These have certain properties of metals as well as polymers, and can also be made to have special properties such as wear resistance, electrical conductivity, corrosion resistance, interference resistance and radiation protection, which have great potential for further development. Bello et al. [22] summarized the fenton-like reagents, mainly introducing the characteristics of conventional fenton reagents while further illustrating the characteristics and applications of three fenton reagents: heterogeneous fenton, fluidized bed fenton and bioelectro-fenton. Alok et al. [23] exhibited various metal species that decompose hydrogen peroxide in fenton-like reactions, paying particular attention to their redox chemistry, practical advantages, describing in details the complex mechanisms, and highlighting the limitations for their environmental applications. Su et al. [24] used Cu-1 IHSs as fenton-like reagents to adsorb and decompose congo red from aqueous solutions based on the unique porous properties of spheres, which demonstrated their potential applications in water treatment. The above mentioned articles indicate the importance of fenton-like catalysts in wastewater treatment. The present review not only reviews the removal of pollutants from wastewater systems by fenton-like reagents, but also explores the thermodynamic as well as kinetics of fenton-like catalyst removal processes.

Carbon-based nanomaterials, which are made of carbon elements compounded with other nanomaterials, have structural units ranging in size from 1 to 100 nm. Carbon nanomaterials have received a lot of attention for their excellent electrical conductivity, mechanical properties, chemical stability and optical properties [25]. Madima et al. [26] showed the mechanisms and applications of carbon-based nanomaterials in adsorptive and photocatalytic removal of organic and inorganic pollutants from wastewater. The potential applications of carbon-based nanomaterials and their derivatives as adsorbents in wastewater remediation were also highlighted. In their review article, Selvaraj et al. [27] presents an assessment of the available literature on adsorption, catalytic degradation and membrane technologies of water treatment relating to carbon nanotube, GO and rGO. The challenges, limitations and future aspects of their feasibility are also discussed for large-scale applications in the water and wastewater treatment industry. However, earlier papers had focused on the desirability of single carbon nanomaterials for wastewater treatment systems. Therefore, this review concentrates on the use of carbon-based nanomaterials as carriers of fenton-like catalysts to investigate whether there is a synergistic effect when the two are combined to improve the efficiency of wastewater treatment.

As shown above, carbon-based materials and fenton-like catalysts have many unique properties and play an irreplaceable role in wastewater treatment processes; therefore, the combination of carbon-based materials and fenton-like catalysts is the focus of this review. This review aims to (1) introduce carbon-based materials such as fullerene, carbon nanotubes and graphene as carriers combined with fenton-like catalysts; (2) present an analysis of the mechanism, preparation and characterization methods of light-fenton, electro-fenton, ultrasound-fenton and microwave-fenton reactions; (3) summarize the effects of carbon-based nano-fenton catalysts on dye wastewater and pharmaceutical wastewater; (4) exhibit the kinetics (pseudo-first-order kinetics, pseudo-second-order kinetics) and thermodynamics (Langmuir, Freundlich) to study the processes of removing pollutants as well as the properties and reusability of fenton-like reagents in wastewater treatment.

## 2. Carbon-Based Nanomaterials

### 2.1. Classification of Carbon Nanomaterials

Nanomaterials refer to materials with a nanoscale greater than 1 nm and less than 100 nm, which have free surfaces and strong or weak interactions between each nano-unit [28]. Due to the particularity of this structure, nanomaterials have some unique effects, including small size effects, quantum effects, surface and interfacial effects [29]. Carbon elements constitute an abundance of carbon materials due to their unique hybridization forms of sp, sp^2^ and sp^3^ [30]. Carbon nanomaterials can be used as catalyst carriers and they can be classified into three categories based on their structures. Those with grain sizes in both directions in the nanometer range are called one-dimensional nanomaterials, such as nanotubes and nanowires. Two-dimensional materials refer to materials where electrons can move freely (planar motion) in only two dimensions on the nanoscale (1–100 nm), for example, graphene. Those materials that have nanoscale scales in all three dimensions are zero-dimensional nanomaterials (e.g., fullerenes and carbon quantum dots) [31,32,33]. 

### 2.2. Application of Carbon Nanomaterials

Carbon nanomaterials can inhibit impurities and defects generated by their variable structure and strong interactions between tubes, layers and sheets [34,35,36]. They exhibit properties in catalytic, mechanical and physical functions (optical, electrical, magnetic and sensitive) that conventional materials do not possess, thus having been applied broadly in the field of nanoscale reinforced fiber composites (Table 1) [37,38].

### 2.3. The Role of Carbon Nanomaterials in Fenton-Like Reagents

Fullerene (Figure 1), also known as soccerene, is an allotrope of carbon besides diamond and graphite, and it has a unique three-dimensional π-electron delocalization structure with strong electron capture and electron transferability under light radiation [50]. Zou et al. [51] doped iron oxide into fullerene to study the degradation of RhB, MO and phenol by introducing visible light in the presence of oxides and found increased catalytic activity, wider pH range and the stability was better than that of loaded fullerene. Meng et al. [52] prepared Fe-fullerene/TiO_2_ with fullerene and titanium (IV) n-butoxide to research its photocatalytic performance for decolorization of methylene blue solution under visible light. The results showed that the fullerene-enhanced TiO_2_ doped with Fe improved the decolorization of methylene blue. Xu et al. [53] have fabricated PHF/hydrous ferrite (PHF/Fh) composites and investigated their photo-fenton activity in simulated sunlight irradiation degradation of acidic red 18. The results revealed that the catalytic activity of PHF/Fh was higher than that of Fh, and the appropriate content of PHF/Fh content could not only improve the stability of Fh, but also increase the production of O_2_ and ·OH. 

Carbon atoms in carbon nanotubes (Figure 2) are dominated by sp^2^ hybridization, while the hexagonal lattice structure has a certain degree of bending, forming a spatial topology. Among carbon atoms, a certain sp^3^ hybrid bond can be formed, that is, the chemical bond formed has both sp^2^ and sp^3^ hybrid states, and these p orbitals overlap each other to form a highly delocalized large π bond outside the carbon nanotube graphene sheet [54]. The large π bond on the outer surface of carbon nanotubes is the chemical basis for the non-covalent binding between carbon nanotubes and some macromolecules with conjugate properties. Yao et al. [25] prepared magnetic metal M (M = Fe, Co, Ni) nanocrystals encapsulated in nitrogen-doped carbon nanotubes using dicyandiamide as a C/N precursor, which was used to degrade dyes. The nanocrystals were found to have high catalytic properties and stability with good reusability. Arshadi et al. [55] degraded methyl orange (MO) by using a non-homogeneous class of fenton reagents immobilized on aluminosilicate nanoparticles and multi-walled carbon nanotubes (Si/Al@Fe/MWCNT) with ferrocene groups. The results demonstrated that the fenton reagents had a good degradation effect on MO and this material had good stability. Samadi et al. [56] prepared magnetic Fe_3_O_4_/MWCNTs composites and then investigated their role as non-homogeneous fenton catalysts in the removal of ciprofloxacin from synthetic wastewater. 

Graphene (Figure 3) is currently the thinnest but toughest conductive nanomaterial. Graphene also shows a large nonlinear anti-magnetism, even greater than that of graphite [57]. Zubir et al. [58] prepared GO/Fe_3_O_4_ fenton-like catalysts for degradation of acid orange 7 (AO7), utilizing GO in transferring electrons to maintain active Fe^2+^ to ensure the decomposition of H_2_O_2_ into ·OH for the degradation of the dye. Zubir et al. [59] also prepared GO and zinc partially substituted magnetite GO-Fe1xZnxOy (0 ≤ x ≤ 0.285) catalysts for degradation of acid orange 7 (AO7). The study revealed that GO enhanced the ability to combine zinc with metal oxide, while zinc limited the crystal growth and formed smaller microcrystal sizes. Liu et al. [60] added GO to ferrite nickel (NiFe_2_O_4_) to obtain GO-doped NiFe_2_O_4_ (GO-NiFe_2_O_4_) for the catalytic degradation of organic dyes under visible light irradiation. It was concluded that GO played a crucial role in the catalytic light-fenton process.

## 3. Classification and Mechanism of Fenton-Like Reagents

### 3.1. Traditional Fenton Reagents

The conventional fenton reagents are a mixed system consisting of H_2_O_2_ and Fe^2+^, which attack organic macromolecules through the ·OH produced by the catalytic disintegration of H_2_O_2_ [61]. In the catalytic degradation process, free radicals break down large-molecule organic matter into small-molecule organic matter or mineralize them into inorganic substances such as carbon dioxide and water, and the strength of free radical oxidation can be determined by the magnitude of the free radical oxidation-reduction potential [62]. It can be seen from Table 2 that the hydroxyl radical has a high oxidation electrode potential and is rapidly oxidized, which leads to the conclusion that hydroxyl radical reaction is a fast reaction. However, fenton reagents have some problems: the utilization rate of H_2_O_2_ is low, and the treated water may have color, which means it is relatively difficult to apply the reagents in drinking water treatment [63]. After continuous exploration and research, it is generally agreed that the fenton reaction mechanism is ·OH generated by a fenton reagent through catalytic decomposition to attack organic molecules, as shown by the following chemical equations:Fe^2+^ + H_2_O_2_ → Fe^3+^ + ·OH + OH^−^(1)
Fe^3+^ + H_2_O_2_ → Fe^2+^ + ·O_2_H + H^+^(2)

### 3.2. Fenton-Like Reactions

Numerous researchers have continuously improved the traditional fenton oxidation method, and a large number of improved fenton reagents have emerged, such as H_2_O_2_/Fe^3+^, H_2_O_2_/O_3_, light-fenton reagents and electro-fenton reagents [64]. Since the reaction principle of the improved technique is still a hydroxyl radical mechanism (Figure 4), which is similar to the fenton reaction, such techniques are collectively referred to as fenton-like reagents methods [65]. 

#### 3.2.1. Light-Fenton Method

Light-fenton method introduces light sources (e.g., UV, visible light) into the fenton reaction to form a UV-fenton system [66]. UV and fenton catalysts can have a synergistic effect on the catalytic decomposition of H_2_O_2_, which has a great improvement on the oxidation of fenton reagents. However, the proportion of UV light in natural light is relatively small (about 4%), and visible light accounts for approximately 43%, thus the introduction of visible light into the fenton system is of great significance. The fenton-like catalyst absorbs photons and oxidizes H_2_O molecules coordinated to Fe(III) to produce hydroxyl radicals [67]. The electrons of iron atoms undergo charge transfer with oxygen atoms, and then the Fe(II) is oxidized to Fe(III) by the dissolved molecular oxygen (O_2_) in solution, or the HOO· and ·OH generated during the reaction [68]. The mechanism diagram is shown in Figure 5, which is represented by the following chemical equations:(3)$$R\text{-}\mathrm{Fe}(\mathrm{III})\;(H_{2}O)\;\overset{\mathrm{hv}}{\to }\;R\text{-}\mathrm{Fe}(\mathrm{II})\;(\mathrm{OH})\;*+H^{+}$$
R-Fe(II) (OH) * + H_2_O → R-Fe(II) (H_2_O) * + ·OH(4)
R-Fe(II) (H_2_O) * + ·OH → R-Fe(III) (H_2_O) + OH^−^(5)

#### 3.2.2. Electro-Fenton Method

The electro-fenton reaction generates H_2_O_2_ or Fe^2+^ through electrolysis in an electrolytic cell, thereby forming fenton reagents [69]. After wastewater flowing into the electrolyzer, the reaction mechanism can be improved under the electrochemical effect. The continuous regeneration of Fe^2+^ at the cathode increases the decomposition rate of organic pollutants thus improving the treatment efficiency of the reagents. The electro-fenton method is to spray oxygen to the cathode of the electrolytic cell to produce H_2_O_2_, which rapidly reacts with Fe^2+^ produced by the oxidation of the Fe anode applied to the solution to generate ·OH and Fe^3+^. This makes use of the strong oxidizing power of ·OH to catalyze the degradation of organic matter [70]. Besides, Fe^3+^ on the cathode can be reduced to Fe^2+^, so that the oxidation reaction cyclically proceeds [71]. The mechanism diagram is shown in Figure 6, and the reaction process is as follows:O_2_ + 2H^+^ + 2e → H_2_O_2_(6)
Fe − 2e → Fe^2+^(7)
H_2_O_2_ + Fe^2+^ → Fe^3+^ + OH^−^(8)
Fe^3+^ + e → Fe^2+^(9)

#### 3.2.3. Ultrasound-Fenton Method

The ultrasonic and vibration of ultrasound can play the role of agitator and mass transfer, and the ultrasound/fenton synergistic method can utilize the cavitation effect of ultrasound radiation to improve degradation rate of pollutants [72]. The mechanism of the ultrasound-fenton method consists of the following two parts: pyrolysis of pollutants due to local high temperature and high pressure that generated by the cavitation of ultrasonic. The strong oxidation potential of hydroxyl radicals generated in the high temperature and high pressure environment has an oxidizing effect on pollutants [73]. 

#### 3.2.4. Microwave-Fenton Method

The mechanism of action of the microwave-fenton method is similar to that of the ultrasound-fenton method [74]. The microwave-fenton method is the utilization of the free radical oxidation and microwaves (thermal and electromagnetic field effects) to decompose pollutants, to promote the decomposition of H_2_O_2_ and the yield of ·OH, and accelerate the polarization of pollutant molecules to achieve synergy with the oxidant, catalyzing destruction of organic matters in wastewater [75].

## 4. Preparation of Carbon-Based Nano Fenton-Like Reagents

### 4.1. Preparation Methods of Carbon Materials

Carbon atoms form many carbon isomers and their bonding is richly varied under high pressure. Carbon nanomaterials have received much attention from researchers due to their excellent properties, such as a wide source of raw materials, controllable structure, good chemical stability and wonderful electrical conductivity, and have shown a great potential in the fields of electronic devices, biology and catalysis. The following are the methods of preparing carbon nanomaterials, and the advantages and limitations of these methods are shown in Table 3.

(1)Method of laser evaporation of graphite, which means that with the help of metal catalysts, pulsed laser bombard on graphite surface to manufacture carbon nanomaterials [76]. For example, Plotnikov et al. [77] fabricated diamond amorphous carbon films with this method.(2)Plasma spray deposition technique means that the plasma-sprayed tungsten electrode (cathode) and copper electrode (anode) are cooled by water. When Ar/He gas carries benzene vapor through the plasma torch, it deposits nano-carbon soot materials on the anode.(3)Condensed electrolytic generation method, with argon playing the role of protective gas, is where the graphite electrode (electrolytic cell as the anode) is employed to electrolyze the molten halide alkali salt and generate various forms of carbon nanomaterials at a temperature of about 600 °C, as well as a certain voltage and current.(4)Graphite arc method is to apply graphite electrodes to discharge in an atmosphere to gather carbon nano-molecular materials from cathode deposits. Xingke et al. [78] employed this method to prepare carbon nanotubes.(5)Chemical vapor deposition is a widely adopted technique for the preparation of carbon materials and can be classified into catalytic and non-catalytic chemical vapor deposition methods. Catalytic chemical vapor deposition is the catalytic decomposition of a gas (or vapor) containing a carbon source as it flows over the surface of the catalyst. Non-catalytic vapor phase deposition requires no catalyst and directly decomposes the gas phase carbon containing organics thermally under a protective atmosphere.

### 4.2. Preparation of Carbon-Based Nano Fenton-Like Catalysts

The fabrication methods of fenton-like catalysts include impregnation, electrostatic spinning, solvothermal synthesis, hydrothermal method, co-precipitation, electrolysis, ion exchange, etc. The impregnation method refers to immerse solid powder or solid of a certain shape and size (carrier or catalyst containing the main body) in a solution of soluble compounds containing active components (main and auxiliary catalytic components) and separating the residual liquid after contacting for a certain period time [79]. Zou et al. [51] immersed Fe_2_O_3_ on fullerenes to form a C60/Fe_2_O_3_ combined with hydrogen peroxide (H_2_O_2_) and visible light to degrade RhB, MO and phenol. 

Electrostatic spinning is a special fiber manufacturing process in which a polymer solution or melt is jet spun in a strong electric field. Under the action of an electric field, the droplet at the tip of the needle changes from a sphere to a cone and extends from the tip of the cone to obtain fibrous filaments [80]. In this way, polymer filaments of nanometer diameter can be produced. Akhi et al. [81] demonstrated that the carbon/MWCNT/Fe_3_O_4_ composite nanomaterials were prepared. Multi-walled carbon nanotubes were synthesized by catalytic chemical vapor deposition in advance, and nitric acid and sulfuric acid were added to make MWCNT undergoes carboxylation reaction, and then FeCl_3_·6H_2_O and FeCl_2_·4H_2_O were respectively added to prepare this composite nanomaterial. The solvothermal method in the original mixture is reacted at a certain temperature and autogenous pressure of the solution, with organic substances or non-water as the solvent [82]. Deng et al. [83] reported the preparation of Fe_3_O_4_-multiwalled carbon nanotube hybrids applying a solvothermal method with acid treatment of MWCNTs and iron acetylacetonate in a mixed solution of ethylene glycol and ultrapure water. 

The hydrothermal method refers to a preparation way in which water as a solvent and the powder is dissolved and recrystallized in a sealed pressure vessel [84]. Xu et al [85]. revealed that the ZnO/GO/SiO_2_ composite catalyst was synthesized by hydrothermal loading of ZnO nanorods on the GO/SiO_2_ surface, and this composite catalyst with H_2_O_2_ constituted a photocatalytic fenton-like system for the degradation of methylene blue. The co-precipitation is the presence of two or more cations in solution, which are present in a homogeneous phase in the solution [86]. Homogeneous precipitation of various components can be obtained by adding a precipitating agent, and it is an excellent approach to prepare ultrafine powders containing two or more metal elements. Song et al. [87] synthesized ultrasmall graphene oxide (usGO) by chemical oxidation of graphite powder and then synthesized Fe_3_O_4_/usGO on the surface of usGO by co-precipitation method.

## 5. Characterization of Carbon-Based Fenton-Like Nanocomposites 

In order to obtain the mechanism as well as the physicochemical properties (e.g., structure, shape, particle size, size distribution, valence and specific surface area) of carbon-based nanocomposites fenton-like reagents, they can be analyzed by some characterization techniques [88]. The main methods described in this article are as follows.

Electron spin resonance (ESR) is the most direct and effective way to detect radicals, and the ESR spectra can give the hyperfine splittings to provide structural information [89]. To verify whether hydroxyl radicals in fenton-like reagents play a role in the degradation of substrates. Peng et al. [90] employed ESR assays to analyze the mechanism because testing 2-propanol was an effective solvent to remove ·OH. Then, by increasing the amount of 2-propanol, the destruction rate of TCS became slower and the removal of TCS was almost completely inhibited, indicating that ·OH is the main radical for TCS degradation in the Fe/H_2_O_2_/GO system. The phase analysis of the crystal material based on the diffraction peak intensity, angular position, relative intensity sequence and diffraction peak shape was carried out by X-ray diffraction (XRD) [88]. The Bragg equation reflects the relationship between the direction of diffraction lines and the crystal structure. For a particular crystal, only the angle of the incident ray satisfying the Bragg equation can produce interference and exhibit diffraction fringes. Zhu et al. [91] performed the XRD analysis on 3D-MoS_2_ nanospheres and GO-loaded 3D-MoS_2_, respectively, and deduced that the preparation of GO-loaded MoS_2_ nanospheres was successful. This material had more surface defects, which was expected to expose more metal active centers, and exhibited an excellent degradation efficiency in AOPs. 

X-ray absorption near edge structure (XANES) is typically a spectrum between the absorption edge at the absorption threshold and about 50 eV above the absorption edge and can be used to provide information about the oxidation state and local atomic environment of the system [92]. Extended x-ray absorption fine structure (EXAFS) is the absorption spectrum beyond the XANES region, ranging from 50 eV above the threshold up to 1000 eV. EXAFS is an element-specific structure characterization technology that can accurately characterize the local environment around the absorbing atom (the type of adjacent atoms, bond length, coordination number, and the degree of disorder in the structure of the specific atomic shell) [93]. 

Transmission electron microscopy (TEM) is capable of observing various defects and atomic structures inside materials [94]. High-resolution transmission electron microscopy (HRTEM) can be extensively implemented for microstructure (including general morphology and lattice images) assessment of materials (especially nanomaterials) and qualitative and quantitative analysis of micro-zone components. [95]. It also can obtain lattice fringe image (reflecting the information of interplanar spacing), structure image and single atom image (representing the atoms or atomic groups in the crystal structure configuration situation) and other higher resolution image information. Yao et al. [96] further revealed the tightly hybridized structure of the catalyst by transmission electron microscopy images, with ZnFe_2_O_4_NPs deposited on the surface of large rGO sheets. Both the edges of rGO and the nanostructures of ZnFe_2_O_4_NPs can be inspected in the TEM images with high resolution. Zubir et al. [97] investigated the structure of GO/Fe_3_O_4_ adopting HRTEM and revealed that the lattice striations of this material were clearly visible, demonstrating the crystallinity nature of the nanomaterials, while it was observed that the relative pressure of the nanocomposites was closely related to the variation of Fe_3_O_4_ nanoparticles.

High-angle annular dark field scanning transmission electron microscopy (HAADF-STEM) can not only distinguish the spatial atomic distribution of the crystal structure, but also directly display the distribution of atomic species and defect structure within the crystal [98]. For strongly correlated materials, the charge density wave, vacancy sequence, ion sequence modulation can be recognized, which are of great significance for understanding microstructure defects and local symmetry breaking. Li et al. [99] obtained HAADF-STEM images and EDS of the same regions in order to study the loading conditions of Bi-FePc on the rGO surface. The EDS element mapping image of Bi-FePc/rGO nanocomposite material demonstrates that the distribution of carbon and iron are consistent, which reported that the Bi-FePc supported on the rGO surface is well dispersed and does not form aggregates. X-ray photoelectron spectroscopy (XPS) allows not only the study of elements and their different valence states on the surface of materials, but also the analysis of information on chemical states, molecular structures, and chemical bonds. In order to better understand the chemical composition and electronic structure of the Fe/C/BN catalyst synthesized before and after the degradation experiment, Yao et al. [100] conducted XPS research, the analysis revealed that the co-doping of B, N and Fe in the carbon skeleton by Fe/C/BN is the main active catalytic center for the effective oxygen reduction in the fenton-like process. 

Nitrogen adsorption is the most common and reliable technique for specific surface and pore size distribution to determine the porosity, surface properties and other parameters of the material [101,102]. Wang et al. [103] investigated the N_2_ adsorption-desorption isotherms of Fe_3_O_4_ nanoparticles and Fe_3_O_4_/GO nanocomposites. The results revealed that Fe_3_O_4_/GO has similar isotherms to Fe_3_O_4_ and type III isotherm according to the classification of IUPAC, which is a typical nonporous material. Fourier transform infrared spectroscopy (FTIR) observes the bond structure of the sample and the interaction in the mixed film, and can obtain the tensile and flexural vibration absorption spectra of the chemical bonds in the sample, which is used for the analysis of the group structure, the qualitative and quantitative analysis of the material [104]. Zhou et al. [105] compared fullerenes and Fe(III)-fullerenes by FT-IR spectroscopy. Original fullerenes are rich in hydroxyl groups and fullerene after loading show lower C–O stretching response and C–O–C and C=O, which revealed that the oxygen-containing functional groups of fullerenes have changed from ·OH to C–O–C and C=O. 

## 6. Application of Carbon-Based Fenton-Like Nanocomposites

The carbon-based nanocomposite fenton reagents have a stable and effective removal function for refractory biological wastewater, toxic wastewater and biological inhibition wastewater. Many factors affect their reaction rate: temperature, acidity and alkalinity, catalyst type, hydrogen peroxide concentration, carrier type, etc. A summary of the applications for fenton-like reagents is shown in Table 4.

### 6.1. Application of Light-Fenton Reagents

The light-fenton method can directly degrade part of organic matter under the action of light, improving the utilization rate of H_2_O_2_, reducing the amount of fenton-like catalyst and further enhance the degree of mineralization of organic matter. Zhou et al. [105] investigated a method to degrade dimethyl phthalate (DMP) based on the high photosensitivity and water solubility of fullerene-enhanced Fe(III)/H_2_O_2_ fenton system under visible light conditions. In addition, the effects of experimental parameters, such as initial pH, iron dosage, fullerene dosage, H_2_O_2_ dosage and light intensity, on DMP degradation were evaluated, and the DMP degradation pathways were analyzed by monitoring the concentration changes of major intermediates. Hybrid nanomaterials formed by nanostructured β-FeOOH were prepared by Fang et al. [106]. The photocatalytic activity of carbon nanotubes/β-FeOOH was examined for the decomposition of MO in the presence of hydrogen peroxide and visible light irradiation, and the photocatalytic oxidation mechanism of the fenton-like system was discussed. The findings illustrated that CNTs/β-FeOOH composites have significant visible light photocatalytic activity than bare β-FeOOH for the oxidation of MO in the aqueous phase. In addition, the growth of visible light absorption intensity and the decrease of β-FeOOH particle size favored the photocatalytic and photo-fenton-like reaction for the destruction of MO. Yu et al. [107] demonstrated Fe_3_O_4_/GO as an efficient photo-fenton catalyst to degrade phenol under UV light irradiation. In this photo-fenton process, the effects of GO content, catalyst dosage, hydrogen peroxide concentration and initial pH of the solution on the degradation of phenol were analyzed. Additionally, Fe_3_O_4_/GO still displayed high catalytic activity as well as good stability after five cycles. 

### 6.2. Application of Electric-Fenton Reagents

The electro-fenton method is highly regarded owing to its in situ hydrogen peroxide production and its ability to effectively control degradation costs [108]. In addition to the oxidation of ·OH in the system, there are also electrochemical anodic oxidation and electrosorption to improve the removal rate. Ai et al. [109] combined copper oxide nanotubes and multi-walled carbon nanotubes with polytetrafluoroethylene to fabricate a composite copper oxide/carbon nanotube/polytetrafluoroethylene cathode and developed a novel electro-fenton system with an oxygen-containing gas diffusion cathode. The electro-fenton system generates copper ions in situ from Cu_2_O nanocubes while electrochemically reducing oxygen to hydrogen peroxide. The fenton reagents further react together to generate hydroxyl groups, which effectively degrade rhodamine B (RhB) at neutral pH, and it would not lose activity after long-term operation. Effect of H_2_O_2_ and Fe^3+^ oxidation on the removal efficiency of anionic (AR14) and cationic (MB) dyes in graphite electrodes modified by electrochemical deposition of GO and rGO by Akerdi et al. [110]. The effects of key parameters—wastewater pH, dye concentration, magnetic nanoparticle loading (Fe_3_O_4_) and current—on dye decontamination were investigated. Furthermore, the chemical behavior of GE-GO and GE-rGO in the oxidation-reduction of MB and AR14 was discussed by cyclic voltammetry. The conclusion was that the removal of AR14 was greater than MB, the removal of rGO-GE system was better than GO-GE, and the magnetic nanoparticles were also effective in generating radical hydroxyl groups in alkaline pH. 

### 6.3. Application of Ultrasonic-Fenton Reagents

The strong oxidation potential hydroxyl radicals generated by the ultrasound-fenton method under high temperature and pressure can degrade organic pollutants in wastewater more efficiently with simple operation and low cost. In the case of ammonium chloride as the nitrogen source, Yang et al. [111] simply carbonized the mixture of glucose and iron salt precursor Fe(NO_3_)_3_·9H_2_O to synthesize Fe/N-C-2 hybrids and then removed tetracycline. The good magnetic properties facilitate the recovery from aqueous media. After six consecutive runs, Fe/N-C-2 maintained more than 88% of its catalytic capacity, which illustrated the high stability of the Fe/N-C-2 mixture in an aqueous solution. Zhang et al. [112] integrated ultrasound and FeGAC to degrade crystal violet. The consequence revealed that the decolorization efficiency decreased with increasing pH, but increased with increasing power density and catalytic dosage. In optimal conditions, the decolorization efficiency reached 88%, and the toxicity decreased sharply at ultrasonic time greater than 50 min. After three consecutive cycles, the decolorization efficiency did not change much and the leaching rate of iron was low, indicating the stability of the catalyst in the ultrasonic-fenton system.

### 6.4. Application of Microwave-Fenton Reagents

The microwave-fenton method combines microwave radiation and fenton reagents to reduce the activation energy of chemical reactions, so that the structure of organic molecules can be more easily destroyed during the reaction [113]. Yang et al. [114] combined microwave technology with fenton reaction for the removal of high-concentration pharmaceutical wastewater. They found that the microwave power was 300 W, the amount of H_2_O_2_ was 1300 mg/L, the reaction time was 6 min, the dosage of Fe_2_(SO_4_)_3_ was 4900 mg/L, the removal rate of COD can reach 57.53%, which demonstrated that microwave can shorten the reaction time and improve its oxidation efficiency. Liu et al. [115] used microwave-enhanced fenton-like treatment of methylene blue in aqueous solution. Operating parameters such as initial pH, MB concentration, and the amount of hydrogen peroxide were investigated to obtain optimal conditions for improving the degradation rate of MB. 

## 7. Studies on Pollutants Removal Processes

### 7.1. Studies on Thermodynamics 

The Langmuir model is based on the assumption of adsorption between gas and solid phases and explains how adsorption occurs on uniform surfaces and it is valid for single-layer adsorption on a limited number of interfaces at the same location. The Langmuir isotherm can also visually describe the adsorption mechanism during the derivation, which lays the foundation for other adsorption models [116]. Freundlich isotherm model is an empirical law obtained when studying the adsorption of gases by porous materials. It applies to liquids and gases. It is a semi-empirical equation, which expresses the non-uniform multilayer adsorption behavior of adsorbates on the surface of adsorbents [117]. Temkin equation considers that when the adsorbent adsorbs the adsorbate in the solution, it interacts with the adsorbed solute and affects the adsorption behavior and process. The specific energy relationship is that the heat of adsorption decreases linearly with the amount of adsorption [118]. The Dubinin-Radushkevich (D-R) isotherm was used to evaluate whether the adsorption process had a physical or chemical mechanism [119]. The equations and descriptions of these adsorption isotherm models are in Table 5.

### 7.2. Studies on Kinetics

The adsorption and desorption rates are mainly determined by the interaction between adsorbent and adsorbate, temperature and pressure. The studies on adsorption kinetics are helpful to explore the mechanism of chemical adsorption and heterogeneous catalysts. The removal kinetic model helps to compare the predicted adsorption parameters with various adsorption behaviors under different experimental conditions. Wang et al. [125] used fenton-like reagents for the oxidation process of Acid Black 1 following the pseudo-first-order kinetics [126]. 

The form of the pseudo-first-order kinetic rate equation is as follows:(10)$$\frac{dC_{e}}{dC_{t}}=k_{p1}\left(C_{e}-C_{t}\right)$$
where *C_e_* (mg/g) and *C_t_* (mg/g) refer to the amount of adsorption at equilibrium and at time *t*, respectively, *K_p_*_1_ is the equilibrium rate constant for pseudo-first-order adsorption.

Equation (1) can be transformed into Equation (2) or Equation (3).
(11)$$\log\left(C_{e}-\;C_{t}\right)=\log C_{e}-\frac{k_{p1}}{2.303}$$
(12)$$\;ln(C_{e}-C_{t})=\ln C_{e}-k_{1}t$$

In order to verify whether the experimental data conforms to the pseudo-first-order rate equation, it is necessary to know the calculated equilibrium adsorption capacity. Even if the adsorption capacity changes slowly, its value is usually less than the equilibrium concentration capacity. Therefore, in many cases, the equation does not match the experimental data in the entire range.

Ifelebuegu et al. [127] investigated the degradation of two EDCs (17β-estradiol and 17α-ethylestradiol) by fenton-like reagents, which conformed to pseudo-second-order kinetics. The pseudo-second-order model is represented by the following equation [128]: (13)$$\frac{dC_{t}}{dt}=-k_{2}C_{t}{}^{2}$$
(14)$$\frac{1}{C_{t}}=\frac{1}{C_{0}}-k_{2}t$$
where *C*_0_ and *C_t_* (mg/L) are the concentrations of solutes at equilibrium and at time t (min), respectively; *k*_2_ (L (mg min)^−1^) is the rate constant.

The typical intra-particle diffusion model is the uniform solid diffusion model, which can describe the mass transfer in amorphous and uniform spheres, and it has been widely used in the analysis of adsorption kinetics. The equation is expressed as [129]:(15)$$q_{t}=k_{id}t^{1/2}+C$$
where *q_t_* is the amount of adsorption at time *t*; *t*^1/2^ is the square root of time; *k_id_* is the intraparticle diffusion rate constant; *C* is a constant.

## 8. Summary and Future Perspectives

This paper reviews the preparation and characterization methods of carbon-based nanocomposite fenton-like reagents along with their mechanisms and applications in light-fenton, electro-fenton, ultrasound-fenton and microwave-fenton reactions. With the continuous advancement of human industrialization, a large number of chemical products are manufactured and employed. These chemicals enter the environment through industrial wastewater and urban sewage, which contains many organic pollutants that are difficult to degrade. If they are not effectively treated, the harm to human society is enormous. Fenton-like reactions are currently recognized as an effective technology for the treatment of refractory organic pollutants, but a further exploration about the mechanism, efficient application and the integration with artificial intelligence still needs to be carried out; therefore, the investigation of fenton-like catalysts is of utmost importance for environmental protection. 

From this review, it is evident that the previous studies have demonstrated a great breadth and depth, but the following aspects still need to be addressed for fenton-like catalysts. Firstly, the mechanism of action of the fenton-like reagent oxidation reactions system should be more clearly understood, it is necessary to strengthen the study on the micro-kinetics, interfacial effects and synergistic effects of external conditions on the catalyst surface. Secondly, research and evaluation of catalyst reaction intermediates are conducted to develop new efficient and inexpensive catalysts, especially those that do not cause secondary pollution and environmentally friendly, which can further improve the utilization rate of ·OH and the biochemical properties of wastewater. Thirdly, the effective applications of fenton-like catalysts are improved based on the full consideration of practicality. From the perspective of practical applications, the main purpose of broadening the use range of fenton systems is to achieve efficient performance of catalysts in neutral conditions. The effect of the new porous catalyst carrier on the reaction active material and its stability ought to be investigated, and they need to be used selectively to improve the activity and enhance the force between the active component and the carrier. Lastly, the development and research of relevant treatment equipment and facilities should be strengthened, and the combination of fenton-like and related technologies should be applied to the treatment of solid and gas wastes, for instance, real-time supervision of water quality and quantity, and link it with the dosage controlling system to achieve the computerized and modernized management, and making the operation simple, saving human resources and improving efficiency. Carbon-based fenton-like nanocomposite catalysts have the characteristics of fast reaction speed, mild reaction conditions and reusability, which will be more widely used in the field of water treatment. It is hoped that this review will provide a valuable reference for future research to gain further understanding and accelerate the development of this field.

## Figures and Tables

**Figure 1 materials-14-02643-f001:**
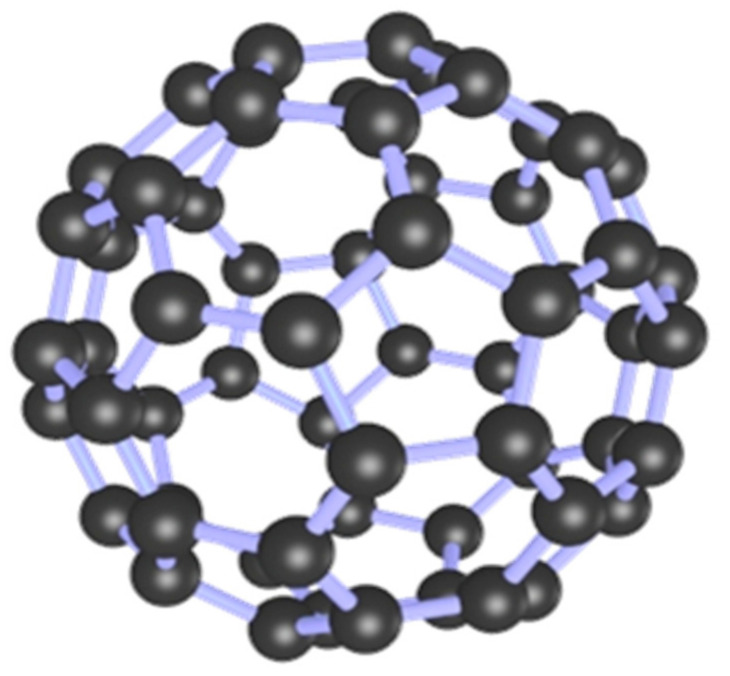
3D structure diagram of fullerene.

**Figure 2 materials-14-02643-f002:**
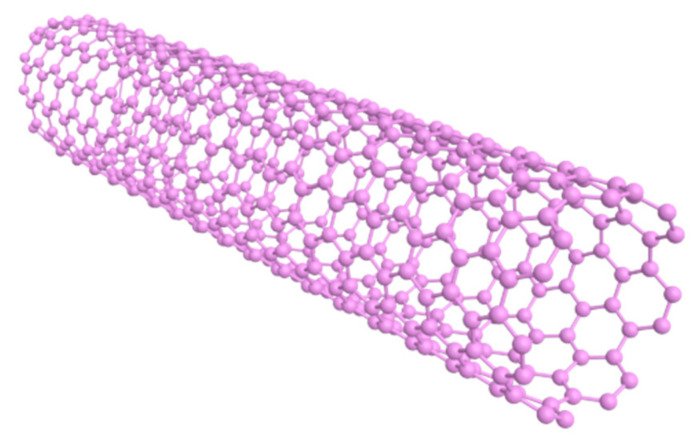
3D structure diagram of carbon nanotubes.

**Figure 3 materials-14-02643-f003:**
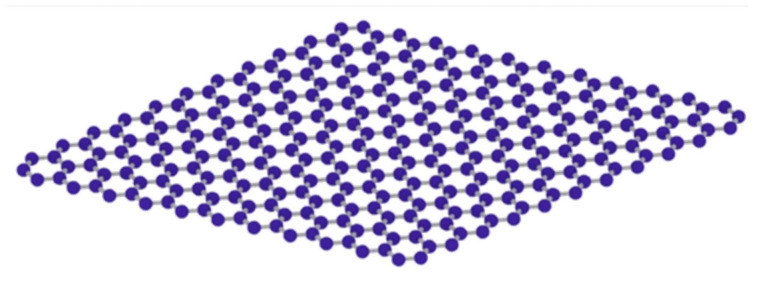
3D structure diagram of graphene.

**Figure 4 materials-14-02643-f004:**
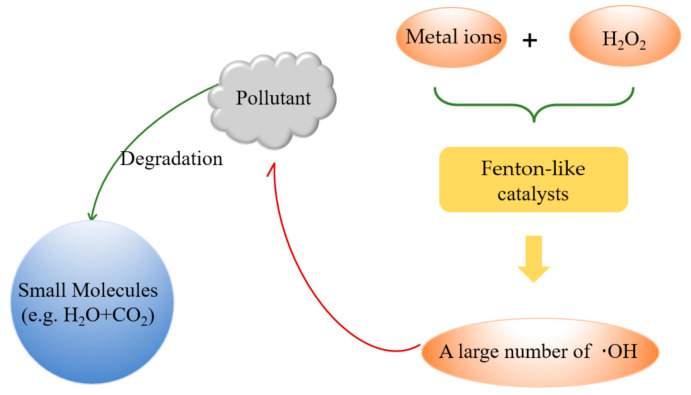
Basic schematic diagram of fenton reactions.

**Figure 5 materials-14-02643-f005:**
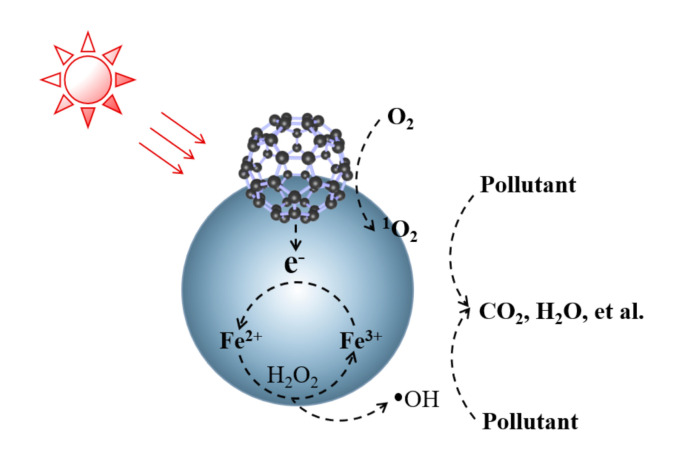
Light-fenton reactions mechanism diagram.

**Figure 6 materials-14-02643-f006:**
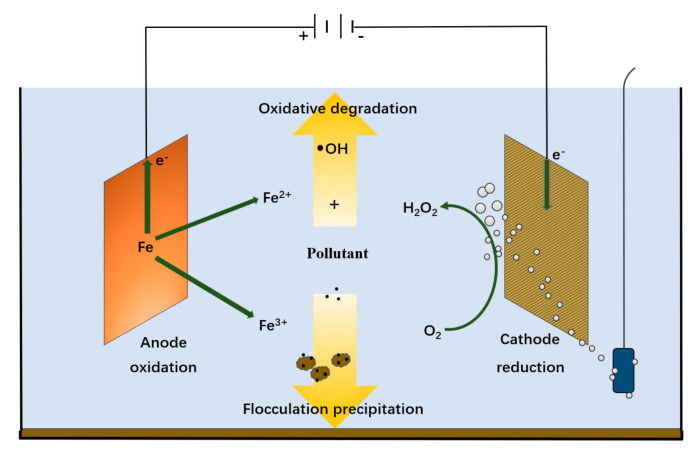
Electrical-fenton reactions mechanism diagram.

**Table 1 materials-14-02643-t001:** Application fields of carbon-based nanomaterials.

Performance	Applications	References
Chemical catalysis	Catalytic catalyst medium	[39,40]
Mechanical properties	The preferred material for high-strength such as reinforcement and toughening	[41]
Magnetic properties	Magnetic absorption, storage memory element materials, soft ferrite materials, etc.	[42]
Electrical performance	Microelectronic device materials, microelectronic device materials, field emission cathode materials	[43]
Optical performance	Large-capacity supercapacitor materials, superconducting materials, nano-integrated circuit materials	[44]
Mechanism performance	Light absorbing materials, optical communication materials, optical recording, optical display, optoelectronic materials	[45]
Thermal properties	Micro-mechanical component materials such as molecular coils and pistons, damping devices and rotary sealing materials	[46]
Physical properties	Micro weapon materials such as micro engines, micro spy vehicles, micro high-efficiency explosives, and materials for aviation and spacecraft	[47]
Sensitive characteristics	Hydrogen storage materials, metal nanowire template materials	[48]
Other	Sensitive materials (sensors, detectors, sensitive electronic scales)	[49]

**Table 2 materials-14-02643-t002:** Oxidation-reduction potentials of various oxidants.

Oxidants	Equations	Oxidation-Reduction Potential (V)
·OH	·OH + H^+^+ e = H_2_O	2.80
O_3_	O_3_+ 2H^+^ + 2e = H_2_O + O_2_	2.07
H_2_O_2_	H_2_O_2_ + 2H^+^ + 2e = 2H_2_O	1.77
MnO_4_^−^	MnO_4_^−^ + 8H^+^ + 5e = Mn^2+^ + 4H_2_O	1.51
ClO_2_	ClO_2_ + e = Cl^−^ + O_2_	1.50
Cl_2_	Cl_2_ + 2e = 2Cl^−^	1.30

**Table 3 materials-14-02643-t003:** Comparison of methods for the preparation of carbon nanomaterials.

Method	Advantages	Limitations
Method of laser evaporation of graphite	High purity of the products	Low output and valuable equipment
Plasma spray deposition technique means	Long electrode life; stable combustion; independent airflow and pressure control; higher efficiency	Expensive carrier gas; small spraying rate; high quality requirements for spraying materials
Graphite arc method	No harmful product formation	Obtain high purity products and consumption of too large amount of energy
Chemical vapor deposition	Simple process, low cost, high yield, suitable for industrial production	Due to the low reaction temperature, the prepared material is defective and requires some post-treatment

**Table 4 materials-14-02643-t004:** Summary of some applications of fenton-like degradation of pollutants.

Pollutant	Materials	Results	References
Dimethyl phthalate	F/fenton; fullerene-Fe(III)/H_2_O_2_ and Fe(III)/H_2_O_2_ fenton systems;	Under visible light conditions, the F/fenton system can almost completely degrade DMP within 50 min, and its large capacity eventually reaches 0.0771 min^−1^, which is 18.5 and 45.4 times higher than that of fullerene-Fe(III)/H_2_O_2_ and Fe(III)/H_2_O_2_, respectively	[105]
MO	CNT/β-FeOOH	The results revealed that the increase of visible light absorption intensity and the decrease of β-FeOOH particle size were favorable to the photocatalytic and photo-fenton reaction degradation.	[106]
Phenol	Fe_3_O_4_-GO	Under optimal conditions (pH 5.0, hydrogen peroxide concentration 10.0 mmol/L, catalyst dose 0.25 g/L), 98.8% of phenol in phenol solution can be removed after 120 min	[107]
Rhodamine B	Cu_2_O/CNTs/PTFE	The degradation of RhB in this E-fenton system reached 80.2% and 89.3% in 120 min at neutral pH and pH of 3, respectively	[109]
AR14; MB	Fe_3_O_4_/GO; Fe_3_O_4_/rGO	In the experiments, the removal of AR14 exceeded that of MB and the rGO-GE system exceeded that of GO-GE, demonstrating that magnetic nanoparticles are also effective in generating free radical hydroxyl groups in alkaline pH.	[110]
Tetracycline	Fe/N−C-2/H_2_O_2_/US system	The maximum removal of TC in this type of fenton system was 92.77%, and the catalytic capacity of Fe/N-C-2 remained above 88% after six consecutive runs, which indicates the high stability of Fe/N-C-2 composites in aqueous solutions.	[111]
Crystal Violet	FeGAC/H_2_O_2_	The optimal conditions for the removal of crystal violet by this class of fenton reagents were an initial pH of 3, a hydrogen peroxide concentration of 1.8 mmol/L, a catalyst loading of 2.5 g/L and a power density of 141 W/L, and a maximum removal rate of 88%.	[112]

**Table 5 materials-14-02643-t005:** Commonly used adsorption isotherm models.

Adsorption Type	Isotherm Models	Description	References
Langmuir	$\frac{C_{e}}{q_{e}}=\frac{1}{k_{L}q_{m}}+\frac{C_{e}}{q_{m}}$	where *C_e_* is the final concentration; *q_e_* is defined as the mole of the adsorbate sorbed per mole of adsorbent at equilibrium; *q_m_* (mg/g) is the maximum adsorption capacity; *K_L_* refers to the constant of the proposed Langmuir model.	[120]
Freundlich	$q_{e}=k_{F}{\left(c_{e}\right)}^{1/n}$	where *C*_e_ defines the equilibrium concentration (mg/L); *q_e_* is the amount of adsorbance (mg/g); *K_F_* (mg/g) and 1/*n* are empirical constants.	[121]
Temkin	$q_{e}=B\;ln(AC_{e})$ B=RT/b	In this equation, *q_e_* is the amount of adsorbance (mg/g); *A* (L/g) represents the binding constant (maximum binding energy); *B* (J/mol) stands for the Temkin constant; *R* is 8.314 J/mol/K and *T* (K) refers to the Kelvin temperature; the constant *b* represents the heat of adsorption.	[122]
D-R	${\ln\;q}_{e}=ln\;q_{m}$ $\varepsilon =RTln(1+\frac{1}{C_{e}})$ $E=\frac{1}{\sqrt{-2\beta }}$	where *q_e_* denotes the amount of adsorbate adsorbed by the adsorbent at equilibrium (mg/g); *q_m_* represents the isothermal saturation capacity (mg/g) and *β* is known as the D-R constant (mol^2^/kJ^2^); *ε* is Polanyi’s potential; the value of *E* gives information about chemical and physical sorption.	[123,124]

## Data Availability

In this paper, the data from Google Scholar, Web of Science, China National Knowledge Infrastructure, China Wan Fang Literature Database, and China WeiPu Literature Database.

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
