# Peer review of "Carbon-Based Nanocomposites as Fenton-Like Catalysts in Wastewater Treatment Applications: A Review"

_materials, 2021, doi:10.3390/ma14102643_

Round 1
Reviewer 1 Report
One of the most serious challenges facing modern humanity is the availability of water resources. The constantly growing population of the planet and the significant development of industry require more and more fresh water. Therefore, a rational approach to industrial wastewater treatment is one of the ways to conserve the planet's water resources. The paper entitled “Carbon-based Nanocomposites as Fenton-like Catalysts in Wastewater Treatment Applications: A Review” presented by Xin et al. examines recent advances in the use of carbon nanocomposites as Fenton-like catalysts in wastewater treatment processes. The review considers the existing Fenton catalysts and their mechanisms of operation in comparison with carbon nanocomposites. The authors considered the main techniques for obtaining composite carbon materials, methods of their characterization and applications. The problems and questions requiring further research in this direction are noted.
In my opinion, the manuscript covers the latest literature in this area and should interest to researchers dealing with the problems of wastewater treatment from various pollutants. I recommend this paper for publication after minor changes.
- I suggest that authors choose other keywords in order to avoid repetition with terms in the title. Keywords should complement the title and enhance search capabilities.
- L. 23 the article is not needed before Water quality
- L. 539 change characterizations to characterization
Author Response
Prof. Jiwei Hu, Ph.D.
1 Guizhou Provincial Key Laboratory for Information Systems of Mountainous Areas and Protection of Ecological Environment, Guizhou Normal University, Guiyang 550001, Guizhou, China
2 Cultivation Base of Guizhou National Key Laboratory of Mountainous Karst Eco-environment, Guizhou Normal University, Guiyang 550001, Guizhou, China
May 13, 2021
Dear Editor,
We are now resubmitting the revised manuscript entitled “Carbon-based Nanocomposites as Fenton-like Catalysts in Wastewater Treatment Applications:A Review ” (Materials -1220415) to your highly respected journal, Materials (Special Issue: Feature Papers in Materials Physics).
We are truly grateful for the critical comments and thoughtful suggestions from the reviewers. Based on these comments and suggestions, we have made careful modifications on the original manuscript and given responses to each of the reviewers’ comments. Revisions made to the manuscript were marked up with the red color. We hope that the revised manuscript will meet the standard of your journal.
Sincerely yours,
Jiwei Hu

Reviewer 2 Report
This paper shows interesting review carbon-based materials such as fullerene, carbon nanotubes and graphene as carriers combined with fenton-like catalysts. There are some issues that need to address:
- Similar reviews have been published recently (e.g., Madima, N., Mishra, S. B., Inamuddin, I., & Mishra, A. K. (2020). Carbon-based nanomaterials for remediation of organic and inorganic pollutants from wastewater. A review. Environmental Chemistry Letters, 18(4), 1169-1191./
Bello, M. M., Raman, A. A. A., & Asghar, A. (2019). A review on approaches for addressing the limitations of Fenton oxidation for recalcitrant wastewater treatment. Process Safety and Environmental Protection, 126, 119-140.
Selvaraj, M., Hai, A., Banat, F., & Haija, M. A. (2020). Application and prospects of carbon nanostructured materials in water treatment: A review. Journal of Water Process Engineering, 33, 100996). It is recommended to add a statement to clearly separate the current work from these similar references and also define the review period (e.g. last five years). Also, prepare statistical data (such as the number of documents, document per country) about you used references by created databank such as Scopus, Google scholar, and web of science.
- Introduction is written simply, most recent research and innovation metal-filled polymer composites performances should be reviewed to show the gap of knowledge. The introduction should be extended with recent research papers. The introduction should be rewritten to show the highlights and novelty of the work.
- The language of the paper needs to be improved, as such; it is really difficult to read...
- I believe this review would benefit from a table that could compare and provide an overview of the discussed approaches. The table should include the advantages and limitations of each approach.
- Section of drawbacks and future could be increased quality of the manuscript.
- The data provided by the tables are low. New columns should be added to the table and more details in each row.
- The manuscript needs an abbreviation list.
Author Response

(The authors gave the same response as above.)

Round 2
Reviewer 2 Report
The comments of my first report have been addressed by the authors.